# Motivating Physical Activity for Individuals with Intellectual Disability through Indoor Bike Cycling and Exergaming

**DOI:** 10.3390/ijerph19052914

**Published:** 2022-03-02

**Authors:** Antonio Martinez-Millana, Henriette Michalsen, Valter Berg, Audny Anke, Santiago Gil Martinez, Miroslav Muzny, Juan Carlos Torrado Vidal, Javier Gomez, Vicente Traver, Letizia Jaccheri, Gunnar Hartvigsen

**Affiliations:** 1Instituto Universitario de Aplicaciones de las Tecnologías de la Información y de las Comunicaciones Avanzadas, Universitat Politècnica de València, 46022 Valencia, Spain; vtraver@itaca.upv.es; 2Faculty of Health Sciences, Department of Clinical Medicine, University of Tromsø—The Arctic University of Norway, 9019 Tromsø, Norway; henriette.michalsen@uit.no (H.M.); vbe005@gmail.com (V.B.); audny.anke@uit.no (A.A.); mmuzny@gmail.com (M.M.); gunnar.hartvigsen@uit.no (G.H.); 3Department of Rehabilitation, University Hospital of North Norway, 9038 Tromsø, Norway; 4Faculty of Health and Sport Sciences, University of Agder, 4879 Grimstad, Norway; santiago.martinez@uia.no; 5Department of Computer Science, Norwegian University of Science and Technology, 7491 Trondheim, Norway; juancarlos.torrado@uib.no (J.C.T.V.); letizia.jaccheri@ntnu.no (L.J.); 6Departamento de Ingeniería Informática, Universidad Autónoma de Madrid, 28049 Madrid, Spain; jg.escribano@uam.es

**Keywords:** intellectual disability, physical activity, mHealth, exergames, gamification

## Abstract

People with intellectual disabilities have more sedentary lifestyles than the general population. Regular physical activity is of both medical and social importance, reducing the risk of cardiovascular disease and promoting functioning in everyday life. Exergames have been envisioned for promoting physical activity; however, most of them are not user-friendly for individuals with intellectual disabilities. In this paper, we report the design, development, and user acceptance of a mobile health solution connected to sensors to motivate physical activity. The system is mounted on an indoor stationary bicycle and an ergometer bike tailored for people with intellectual disabilities. The development process involved the application of user-centered design principles to customize the system for this group. The system was pilot-tested in an institutional house involving six end-users (intervention group) and demonstrated/self-tested to relatives of persons with ID and staff (supervision group). A System Usability Scale and open-ended interview in the supervision group were used to assess the user acceptance and perceived usefulness. Results indicate that the users with an intellectual disability enjoyed using the system, and that respondents believed it was a useful tool to promote physical activity for the users at the institution. The results of this study provide valuable information on beneficial technological interventions to promote regular physical activity for individuals with intellectual disabilities.

## 1. Introduction

Intellectual disability (ID) is characterized by “significant limitations both in intellectual functioning and in adaptive behavior as expressed in social and practical skills. This disability originates before the age of 18” [1,2]. ID differs greatly from person to person, but it is always associated with impaired cognitive abilities. The term “cognitive abilities” means the ability to perceive, process, remember, consider, retrieve, and act purposefully to the information in the environment [3]. Compared to the general population, persons with ID are at an increased risk of health problems [4], have lower perceived health [5], and have difficulties finding appropriate health care [6,7]. Specifically, persons with ID have lower levels of physical activity than the general adult population [8], low scores on physical capability tests [5], and a higher incidence of obesity [9]. Haveman et al. estimated that 50% of people with ID have a sedentary lifestyle, and 40% of them have low levels of physical activity [10]. A previous study revealed that only 7% of males and 8% of females with Down syndrome met the recommendation of 30 min of physical activity per day [11]. A review by Dairo et al. [12] found that 9% of the individuals with ID worldwide were able to meet the WHO’s recommendation of minimal physical activity.

Among the reasons for these low activity levels, we can find several barriers for individuals with ID to participate in physical activity: lack of resources for necessary support, reduced physical and behavioral skills, and lack of available programs [13]. As a solution to these problems, it is suggested to investigate successful methods for encouraging physical activity for individuals with ID [14], suggesting a better use of theory drawn from intervention designs in community-based settings [15].

Technology can support people in the self-management of chronic conditions [16,17,18]. Health-related video games are effective for shifting behavioral changes and promoting health by influencing health-determining activity [19]. Active video games, also known as exergames, have been investigated and found to be promising for individuals with ID [20,21]. Before these solutions can be implemented outside of the lab, they must first meet the users’ needs, which can be ensured by systematically analyzing user preferences [22]. Using touch-screen devices such as smart-phones, tablets, and iPads has been proven to have low cognitive demands and could be used to improve commitment to physical activity [23,24].

The main research questions of the study were twofold: “*(1) How can we develop exergames for people with ID?*” and “*(2) How do these customized exergames work for people with ID?*” By answering these research questions, we aimed to discover how to make a system to promote physical activity that is appropriately designed to be used by individuals with ID. To this end, User-Centered Design principles were used to understand user preferences and perceptions, and thereafter to create a prototype of the system. The final stage of the study included a pilot test with three people with ID and a demonstration/self-test to relatives of persons with ID and staff.

## 2. Related Work

Several studies have identified barriers that cause individuals with ID to have low levels of physical activity [25]. Lack of motivation is emphasized as one of the reasons for individuals with ID not being physically active [26], and is related to the fact that they do not understand the benefits of exercise. Other barriers that have been pointed out include a lack of options for physical activity and programs aimed at individuals with ID [27], or that physical activities are too difficult or boring. Additionally, the preparation, skills, and motivation of the staff working at institutions or day care centers for persons with ID have been demonstrated to have a positive impact [28].

In addition to revealing these barriers, the studies also made suggestions on how to overcome them. Van Schijndel-Speet et al. [29] stated in their study that it would be beneficial to increase staff knowledge on physical activity and available options for physical activity and materials. Being rewarded and praised for performance in the form of feedback, medals, or awards has been proven to be a promising way to create interest in physical activity for individuals with ID [30].

Exergames, also known as active video games (AVG), are defined as video games that require body movements to control the game [31]. In contrast to regular video games, exergaming promotes both exercise and video gaming at the same time [19]. Several terms are used for exergames, such as “active video games” and “interactive video games”, and they are also defined as “any type of video games/multimedia interactions that require the game player to move physically” [32]. Exergames have been involved in several intervention studies that investigated the health-related benefits of using exergames [33,34]. The objectives for using exergames in the recent scientific literature are heterogeneous; while some studies looked at increasing physical activity [35,36,37,38], other studies were focused on motor coordination control [39,40]. Some of these studies implemented a form of user testing. The most thorough testing was conducted by Davison et al. [36], who they were able to test an exercise program involving exergames on over 109 students for a year. However, these test participants had developmental disorders that do not come under the qualification of ID, which weakens the relevance of the results for this study. Another interesting study, executed by Chang et al. [35], tested exergames on two individuals with ID in ten series of three minute sessions over five days. They concluded that the intervention had a positive effect on increasing physical activity. Their methodology and design had many similarities to the present study, such as the use of an ergometer bike and an entertainment system to give the rewards. A more recent study on the influence of exergames evaluated the improvement in both the psychological and the physiological benefits of exercise in children with mild ID using Augmented Reality, showing positive impacts as expressed by preferences for future gameplay [41].

Most of the interventions had a positive effect in persons with either developmental disabilities, autism spectrum disorder, or ID when the intervention included a form of exergame or technology. In this study, we aimed to design, develop, and investigate user acceptance of a mobile health solution to motivate physical activity in adults with intellectual disability.

## 3. Methods

This study aimed to develop and test a combination of hardware and software for promoting autonomous exercise in persons with ID. As mentioned, ID differs from person to person and categorization is made depending on the degree of severity. In this study, we focused on people with mild to moderate ID, which is characterized by a slower ability to learn new information or skills but allows independent living with the right support, or living in apartments attached to services [42]. Reasons for choosing this target population were that it is the most prevalent group and that it is the group of persons who can benefit the most from technology-based interventions [20,42].

The proposed methodology consists of three subsequent stages which can be executed in cycles (iterations). This paper reports the results of one iteration. The first phase involves the definition of requirements and involves a comprehensive study of the literature and current existing exergames, as well as interviews with experts, persons with ID, and their relatives. Based on these requirements, Phase 2 involves the development of the exergame and the verification of its proper functioning in laboratory tests. Once the system is ready and all the design requirements are fulfilled, the system is tested in Phase 3, involving real users with ID who test the exergame. This latter stage includes evaluation of the usability of the system and feedback collection by means of open interviews with relatives of persons with ID and staff members from the day care center.

Based on previous works [20,43], an indoor bike cycling system was chosen as the principal product for performing exercises, so that persons who have a mild condition can use the system by themselves and those who do not can receive support from parents or staff members at an institution, who can guide them during the exercise sessions. The method for developing this study was based on the design paradigm of computer science, as defined by Hevner [44]. This paradigm requires the analysis of requirements and specifications as the first stage, and subsequently the design and implementation of the system followed by testing. The hypothesis of the study was that as persons with ID enjoy watching videos and movies, in addition to playing video games, these tools can be used to create an exergame to increase the performance of physical activity.

### 3.1. Design Requirements

Literature is scarce on the use of exergames for promoting physical activity in persons with ID. Most reported interventions rely on existing videogaming technology such as the xBox Kinect [45], Nintendo Wii [46], or custom hardware to implement friendly environments in which the persons with ID can practice and develop motor skills (performance and acceleration) and maintain physical activity routines. A few of the published studies include a description of the system design. The vast majority included assistance features (voice, pictures, schemas, etc.), real-time feedback about the level of performance, and other features for connecting with peers and socialization.

With respect to current solutions, there are currently several applications which can be linked to static bikes and other motion sensors that enable tracking of physical activity (duration and intensity). Static bikes are equipped with a built-in screen to show entertainment to the user as he or she performs the practice. However, these solutions are not tailored to persons with ID. To the best of our knowledge, the only solution available is the FunDoRoo app, which contains an activity plan and instructions on how to do several fitness activities. As the game uses text to describe most actions and information, it does not suit individuals with more severe degrees of ID, since most of them are not able to read long texts.

The first phase of the methodology included interviews and workshops with experts, persons with ID, and relatives. To this end, seminars were organized with healthcare professionals, psychologists, professionals from day-care centers, and experts in software for supporting persons with ID. Individuals with ID often require support and guidance from parents or staff members when doing activities. It is expected that when using this system, parents and staff members will have to provide guidance and support for the user to tell them how to use it and provide motivation to use it. Stakeholders emphasized the importance of predictability and that the system should express in a plain and engaging manner what is happening. A system capable of showing both full text, short text, and audio for explaining elements was defined as a success factor for the system. Table 1 summarizes the key design requirements for the system (Table 2).

### 3.2. The Exergame Design and Development

The application development was done in Unity. Unity Engine provides different approaches for game development. This platform supports both 3D and 2D modes for design. The exergame was designed to implement both modes in the app. The 2D design mode was used for navigation menus, status view, and displaying video mode option during an exercise session. The 3D design mode was required to create the game mode, which allows navigating in a virtual environment. As the game mode was not implemented, the current version of the app is only in 2D. Additionally, Android Studio was used for low-level communications with the system. Android native libraries are compatible with a wide range of low-cost tablets and smartphones. Unity executable software is compatible with Android and iOS operative systems.

The exergame was designed to run on a tablet or a mobile phone, so the user can use it independently. The system design included two proposed ways for tracking physical activity on an indoor bike. The first solution used a regular outdoor bike mounted on a roller stative that keeps the bike steady and provides a resistance on the power wheel without moving. The materials for this first solution were a Tacx Flow Smart trainer that supports Bluetooth Low Energy and Ant+ connection. This trainer measures speed, cadence, and resistance, and it is possible to adjust the resistance on the power wheel. Cadence is a standard unit of measurement for bike trainers, and is defined as the frequency of the pedal when cycling.

The second solution used an ergometer bike of any type mounted with a cadence sensor on the pedal. An U.N.O. Fitness ET1000 ergometer bike mounted with a sensor in the crank was used to register the cycling activity and transfer it to the control unit. Using a separate sensor made the system less dependent on the type of ergometer bike, and made the set-up compatible with any ergometer bike at home or in a day-care institution.

The Wahoo cadence sensor, which supports Bluetooth Low Energy (BLE), was chosen for building up the system. BLE was launched in 2010 through Bluetooth 4.0 and is supported by most iPads and tablets produced after 2012. The sensor uses the FTMS protocol through BLE, the same as the Tacx Smart Flow trainer, which makes the connection implementation simpler as it can be used in both solutions.

Individuals with ID often require support and guidance from parents or staff members when doing activities. It is expected that parents and staff members will have to provide guidance and support for the user to tell him or her how to use it. Functional and nonfunctional requirements were extracted from two illustrative use cases of the system (Table 1).

### 3.3. User Acceptance Tests

The System Usability Scale (SUS) [47] was used for assessing the perceived usability of the system. SUS is a ten-item scale used to measure industrial usability. Each item constitutes a statement of the user’s opinion of the system, where the response indicates the degree of agreement or disagreement with a statement on a five-point scale. The results of the SUS questionnaire are aggregated on a scale of 100%, in which a score of 100% indicates that the system is likely to be accepted in all situations. The statements on the scale are set up such that the half of them would commonly lead to strong agreement and the other half to strong disagreement. SUS has been proven to be a reliable tool for measuring perceived usability [48]. The SUS questionnaire was complemented with an open-ended interview used to assess the estimated time used on routines of the exergame and the occurrences of errors during their execution. The ISO 9241-11 standard for evaluating usability was used for the interview, covering aspects related to effectiveness, efficiency, and satisfaction. In addition to the SUS scale, questions were asked about the users’ technical experience, features missed, and opinions. The SUS questionnaire and the interview were only used in the supervision group (relatives and staff).

### 3.4. Ethical Considerations

All the users who participated in the study gave their written consent. For the people with ID (intervention group), the closest family member of the participant with ID was authorized as a decision maker for the actual test user and gave their informed consent. The consent agreement gave information about what participation involved, that it was completely voluntary, what would happen to the information gathered, and who was responsible. To ensure the anonymity of the participants in the test, information such as age, type of ID, gender, and other characteristics was not included in the results. Information about the results was obtained by interviewing the employee that had been responsible for supervising the tests for the users. The results of the quantitative analysis based on SUS were extracted from the answers of the supervision group (relatives of persons with ID other than those in the intervention group) and staff from the day-care center who gave their informed consent to participate in the study.

## 4. Results

The developed exergame was tested in a day-care institution that arranges different activities for individuals with ID. Three users at the day-care institution tested the system in sessions spread out on four days, guided by the staff. One of the members in the staff was the contact point for receiving information on how to operate the system and whom to contact for helpdesk and support. This staff member was also responsible for providing information on how the tests had gone in an interview afterwards. This section describes the implementation and technical features of the system and the quantitative and qualitative results of the tests performed with the users, relatives, and staff.

### 4.1. A Static-Bike-Based Exergame

The final deployment of the exergame system is described in Figure 1. The cadence and the speed of pedaling were acquired by means of a Wahoo cadence sensor (right side of the Figure 2) and the Tacx Smart Flow (left side of Figure 2). These two sensors transmit values using the broadcast channel of BT LE so the app in the tablet can read them for further processing.

The app was design for use with a touchscreen or a tablet. When designing for a touchscreen, all navigation and action that requires user input must be accessible through the display view, and all interactive elements must be easily identifiable the target users of the exergame (Figure 3). The virtual environment of the exergame is controlled by a main application developed in Visual Studio. The application tracks the navigation, changes in the settings, and data storage, manages Bluetooth communications, and maintains the application life cycle.

The app allows the user to customize a profile and track the sessions he or she performs (duration, intensity, goals, achievements, etc.). The main menu of the application shows navigation menus with which the user can select the type of routine and the entertainment modality (ride in the city, ride in the countryside, music, TV show, or a movie). A Unity application is usually organized into scenes, each of which stands as a container for an environment that has common properties. Several scenes are usually used to separate different levels in a game, where each level differs significantly in visual content and functionality. In the implementation of the exergame, a unique scene is used with several components that build up the functionality of the app. The core of the app consists of handlers managing the state of the game and panels displaying the app content.

### 4.2. Acceptance Tests

After the deployment of the system in the day-care institution, the system was present to the expert group in a meeting. A live demonstration was done with the app running on a monitor and the author going through all the features of the system. Afterward, there was a discussion of the system and then the attendants completed a SUS scale about the system.

The test phase was set up to last four days, Monday to Thursday, where the system was at the day-care institution’s disposal all the time. The system was set up in an activity room at the center which already contained two other ergometer bikes and a treadmill. The staff presented the system to the users who had agreed to participate and oversaw the execution of test sessions. If any of the users showed signs of disliking it, they could stop and did not have to do it anymore.

The system was tested using the three-wheel bike and the Tacx Smart Flow trainer. A simple user manual was sent along with the system to explain how to use it. The system was set up with a Samsung Galaxy Tab A 10.1 with the app installed. The app displayed five different videos to the users. Two of the videos were recorded bike trips through the city center of Tromsø (Norway), which were three and eight minutes long; two of them were Mr. Bean cartoons that were each eleven minutes long; and the last was a bike trip video through Swiss Alps lasting 14 min, extracted from the Tacx videos with permission from the Tacx Inc. company.

The intervention group tested the system by themselves with the support of staff. After the test phases, the employee was interviewed about how the testing had gone from an objective point of view. The contact person was asked to give an opinion on the system based on their observations of the system in use.

Additionally, the system was demonstrated and self-tested in a specific session with relatives of persons with ID and staff members (supervision group). Figure 4 shows the scores of the SUS questionnaire for each of the members in the supervision group and the dispersion of the individual SUS items. The aggregated score was SUS = 70.83 (16.86) (mean (standard deviation)) with a Cronbach’s alpha of 89.93% (k = 10), which denoted an acceptable consensus between respondents.

The diversity among the calculated scores was high (Figure 4), and a reason for this could be the diversity and perceptions of IDs. The severity degree of ID is divided into mild, moderate, severe, and profound based on the abilities an individual has, measured with IQ scores and other ability tests. It is likely that among the parents and institution staff, their personal experience is related to people with different degrees of ID or other diverse personal characteristics, and the answers will therefore vary accordingly.

When looking at the scores of the different statements on the SUS scale, there were two that were separated by significantly lower ratings. These were due to the statement which asked if a user with ID would require support from personnel to use the system. This might have been because of the more advanced settings, such as setting a speed limit or constructing music playlists, where it is likely that users would require help. Starting an exercise session without editing settings seems to be more legitimate, as statements such as nine had a high score. Statement nine states that a user is confident in using the system alone. The low score on statement ten is related to how much a user needs to learn to operate the system. It is reasonable to think that a user would need to be told how to use the system before engaging with it, and that the score is relatable to the score on statement four.

### 4.3. General Feedback

The users involved in the testing were experienced users of tablets and often used online broadcasting platforms to watch their favorite videos. The users were also experienced with playing games on smartphones or tablets and, at the institution, they have scheduled periods for video games or surfing the internet. In general, all users enjoyed the system; however, there were different experiences during the intervention. All users enjoyed watching the videos that were available in the app. However, when biking through the city center, they did not show the same enthusiasm as for other entertainment options. The entertainment videos, such as the cartoon, appeared to be more interesting. Two of the users watched the cartoon videos more than three times, while one watched them twice.

One user was not familiar with how to use a bike and did not know how to operate the pedals, which would lead to the playing of a video. After being instructed by the staff member, this user eventually learned how to use the pedals to cycle. Another user watched all the videos two times during exercise sessions and then said it was not interesting anymore. This user had a wish for more cartoon videos and videos that lasted longer, e.g., 20–30 min. Another user also watched all the videos during sessions and said that it was interesting, but was confused by the fact that he could be biking through the city center at the institution’s activity room. The system was able to start an exercise session within an estimated range of 3–5 s, and except for the user that did not know how to cycle, there were no problems and all users were able to start an exercise session by themselves.

Qualitative feedback from the meeting stated that the system was promising as it is, but also suggested interesting improvements. One of the proposals was to add a scoreboard based on distances achieved in training sessions, where it would be possible to compete against other users of the app. One of the members at the meeting who worked at an institution said that this was likely to make the users engage more, as their members enjoyed competing against each other. Another wish was the possibility to watch longer videos such as movies, where you would be able to start watching a movie during one session and continue in another exercise session.

## 5. Discussion

In this paper, we present and describe an exergame system for motivating people with ID to improve their levels of physical activity. People with ID are at a higher risk of sedentarism, overweight, and developing cardiometabolic diseases [49]. The system was designed based on user-centered design principles, which led to the definition of a system that could make users feel comfortable and motivated to perform physical activity. The choice of a bike was grounded in the benefits that moderate-intensity and aerobic activity have on health and fitness indicators. The proposed system is based on two approaches, with a built-in ergometer and a cadence sensor connected through Bluetooth LE to a central unit, which can show videos (entertainment or landscapes) according to the intensity of the activity the user performs.

The system was tested in a day-care center involving three users with ID (intervention group) and demonstrated/self-tested to a group of five relatives and a staff member (supervision group). The results from the testing at the day-care institution were provided through an interview with one of the staff members and the SUS questionnaire was applied to assess the system’s usability and perceived usefulness. One person’s opinions are a small basis on which to assess the system and evaluate how well it achieved the study goals. However, the interviewee had long experience with individuals with ID through working with them for a long time and the results can be considered to have significant relevance as test results. The presentation of the implemented prototype for the reference group improved the foundation for assessing the system and improved the quality of the SUS scores.

The SUS score results imply that the system achieved almost acceptable usability for individuals with ID, based on the perceptions of the supervision group. The significant differences between the scores calculated for each SUS form are an indication that the app could be highly acceptable for some individuals with ID. This was also the observation that the staff member at the day-care institution reported. One of the test users in the intervention group understood straight away how to use the system, while the two others needed a little more assistance. Persons with ID often receive assistance from institution staff or family members when performing different activities. Taking this fact into consideration in the assessment of the usability justifies the statement that the system is user-friendly for individuals with ID, despite receiving low SUS scores in some of the questions from the supervision group.

Users spotted limitations in the entertainment offered by the system. Therefore, it could be considered to make a more advanced system and include more features in the app. Some of the barriers for individuals with ID in engaging in physical activity are a lack of motivation and feeling bored during activities. Because of these types of barriers, the motivation for individuals to learn how to use this app is presumably low, as it requires the user to do physical activity. Other approaches based on fully immersive environments have shown positive results and should be explored [41]. Design choices that give reasons for not using the system, such as including more advanced features, were therefore avoided to the extent that it was possible.

Using amusing and entertaining videos as a reward proved to be promising for encouraging physical activity on a stationary bike among the users in the test. All the test users enjoyed watching cartoon videos while cycling, and two of them also found the rides to the city center entertaining to watch. Making more videos available is therefore likely to improve the effect of the system, which has always been a point with the design. The entertainment content available in the system should be dynamic and customizable to adapt to the individual preferences among the user group.

One of the users that tested the system did not understand how the display could show videos of the city center when cycling in the activity room of the institution. However, the user enjoyed using the system when watching the cartoon videos while cycling and had no problems with them. Playing videos immediately after the pedaling gave a clear correlation between the exercise performed and the reward received. There was a user who did not understand that pedaling was needed to run the video. However, the user did eventually discover how it worked by receiving some initial instructions from the staff member. These results show the advantages of keeping the rewards from being too abstract, as recommended by the experts.

The results of the tests indicate that the system has a user-friendly UI for individuals with ID, which was an important goal of this study. The promising results from the testing indicate that adapting aspects such as usage of symbols, short text, and audio commands is a justified decision. Another important aspect for improving the user-friendliness was the automatic set up of the connection over Bluetooth. The app searches for an available device that supports the proper type of bike data and connects to it. This procedure prevents the user having to navigate in the UI to find the correct device in a list and thus simplifies the usage of the system.

This study created a prototype that combines measurement of bike activity data and an app that receives the data in an entertainment system. Results from usability testing of the system indicate that the system made physical activity on a stationary bike more amusing for individuals with ID. Through previous knowledge on behavioral change in individuals with ID and studies on the use of technology, we learned that many individuals with ID enjoy watching videos such as cartoons and TV shows and that they often use tablets or iPads. Additionally, the rewards that this system gives must be given immediately as the physical activity is conducted. The interest and preferences among individuals with ID differ immensely from individual to individual. Therefore, the content that is provided in the app should be dynamic to suit the different users. The application within the system was developed with the Unity Game Engine since it is a convenient tool to make such types of app (visual and interactive) and it supports cross-platform compilation. The system can be installed with a Tacx Smart Flow trainer, a regular bike, or an ergometer bike of any type and a Wahoo Cadence sensor.

We learned that many individuals with ID enjoy riding their bikes outside when the weather is nice in summertime. However, nice weather and summer are not common in places like northern Norway, and the number of bike rides possible during a year is limited. Therefore, to take advantage of the fact that many individuals with ID enjoy bike cycling, this system uses a stationary indoor bike mounted with capable sensors. We wanted to be able to use a regular bike in the system in order to make the connection to outdoor biking clearer. However, when visiting the day-care institution, we learned that several of the users liked to use the ergometer bikes in the activity room, which justified having this as an option as well.

There have been recent studies implying a need for more technological interventions with individuals with ID to improve their health conditions [40,50], but the literature is scarce on the promotion of physical activity to individuals with ID with use of technology. Chang et al. [35] used an ergometer bike along with an entertainment system to encourage individuals with ID to cycle, but without measuring the duration and intensity of the activity. With sensors measuring cadence or speed, as in the proposed system, a better quantification of the performed exercise is provided, and it is possible to adjust the effort needed to yield a reward. Additionally, the system [28] required an external mini-computer and an external monitor to control the entertainment system, in comparison to using a tablet or iPad, which many of the users possess already. In the entertainment system, they provided a single video for each user. This makes it a more limited solution compared to the system in this study, which provides a list of videos that can be edited to suit the user.

Promoting physical activity among individuals with ID is often a battle for parents or institution staff [51]. They struggle to get their child or institution member interested in doing physical activity, as they would rather take part in sedentary activities such as watching videos or surfing the internet. Making systems like this provides them with a potentially helpful tool in the battle against a sedentary lifestyle amongst individuals with intellectual disability. As research has shown, individuals with ID are not the only group struggling with a sedentary lifestyle, and this system may also function well for other groups that are unmotivated to be physically active.

If the system does make individuals with intellectual disability engage in more physical activity, it could also contribute to decreasing the risk of diseases and health-related problems that often follow a sedentary lifestyle. Municipal services could adopt this system as part of a larger activity program to promote a healthy lifestyle for individuals with intellectual disabilities. This study also showed a procedure for developing technical solutions for individuals with intellectual disabilities, with good indications that it will yield a successful solution. The study provides valuable information for research involving intervention studies on individuals with intellectual disability and technical solutions to encourage physical activity.

### 5.1. Limitations

This study is subject to some limitations, which are listed in this subsection. The user tests were conducted for the purpose of usability and perceived usefulness, providing only brief indications on how the application could function over a more extended period. The enthusiasm shown during testing may have come from the fact that the concept was new and exciting, but over time the enthusiasm might wear off. Research shows that it may take up to six months to make a habit of a new behavior, such as using this application [52]. Therefore, an extensive test should last 3–6 months and have a suitable user with ID making use of the system in a realistic environment, e.g., at home or in an institution.

Results of the tests from the day-care institution were gathered through an interview with one of the staff members. Having an objective assessment means that the quality of the results relies on the interpretation of the interviewee and adds the risk of influence of biased opinions. When using an interview to collect the results, an interviewee might feel obliged to answer kindly and politely, causing the results to be more positive than when using an anonymous questionnaire. The interview answers were written notes, which can also obscure results and increase the risk of bias when the notes are interpreted.

Other limitations with the testing are the extent and comprehensiveness with which they were conducted. The tests that were performed at the day-care center were done over four days. The positive result could have been influenced by the excitement for new interesting equipment, but as the system is explored excitement may wear off. Having a longer test period would reveal with more certainty the usefulness of the product. Having more test users would also increase the probability of revealing more lackluster responses and failures with the system.

The statements in the SUS scale had to be altered into objective opinions about the usage for individuals with ID and translated into Norwegian. This altering of the statements distinguishes them from the original model and the relevant research conducted around it.

Despite all the limitations of the testing that was conducted, the results still have credibility to the extent that they should be included in the assessment of the system. All the opinions gathered were from people with broad experience with individuals with ID, as they have interacted with individuals with ID over long periods of time (years). Nielsen [53] used the findings in the study “A mathematical model of the finding of usability problems” to claim that five users will be enough to find 85% of the usability problems. Nielsen also stated that 15% of hidden problems remaining are likely to be found by doing another test with five new users. The SUS scores gathered in the present evaluation came from six forms, which means that they should represent a significant portion of the problems with the system. The translation of the SUS statements was done carefully and in dialogue with professors within the computer science field. We believe that much of the relevance for each statement on usability was maintained sufficiently to use the statements in this evaluation.

### 5.2. Future Work

Implementing new features such as multiple languages, responsive resistance in the ergometer, and other multimedia will extend the possibilities and improve the probability that the system will be a success over the longer term. Integrating online video platforms and adding a scoreboard will be important aspects for maintaining user interest in the product over time. Adding a game mode with dynamic content provided through a server will provide features that meet more interests and increase the potential for users of the system.

It is worthwhile mentioning that when implementing these new features, it is necessary to keep in mind that the usability for users with ID needs to be maintained. As we observed, the SUS scores gave indications that the usability prototype was assessed to be demanding for some users with ID. To achieve acceptable usability, it is recommended to continue with a close user integration in the development process, such as running usability tests and having reference group meetings. A possibility is to make the layout more configurable in a way that allows the user to choose how advanced the information presentation is. More voice commands should be added for actions and activity statuses, but muting them should be easy as they may become annoying over time. For instance, in the beginning, the voice commands help the user to understand the system, but after the user is familiar with the system, they are no longer required.

## 6. Conclusions

We showed that close user interaction is valuable for developing an exergame for individuals with intellectual disabilities. We devised the system requirements through research using relevant studies and reports, experts with knowledge and experience, and user engagement observations from parents and institution staff. Successfully implementing almost all the requirements for a functional product resulted in a promising solution that was assessed to be user-friendly for individuals with intellectual disabilities.

We showed that setting up a stationary bike connected to an entertainment system was an effective way to encourage physical activity. The prominent success factor was the use of entertainment videos as an immediate reward to motivate the user to perform cycling activity. The app that includes this motivation system allows the user to keep track of the activity conducted and to be motivated to complete a weekly goal of activity time. Another important factor was to make the app configurable in content to adapt to the many differences that appear in the group of individuals with intellectual disabilities. The most prominent upgrade after getting the test results is to allow support staff to add videos dynamically, which is part of the ideal design of the system.

## Figures and Tables

**Figure 1 ijerph-19-02914-f001:**
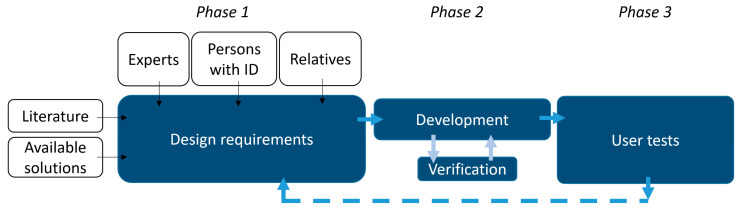
Proposed methodology for the definition of requirements, implementation of the exergame, and user testing.

**Figure 2 ijerph-19-02914-f002:**
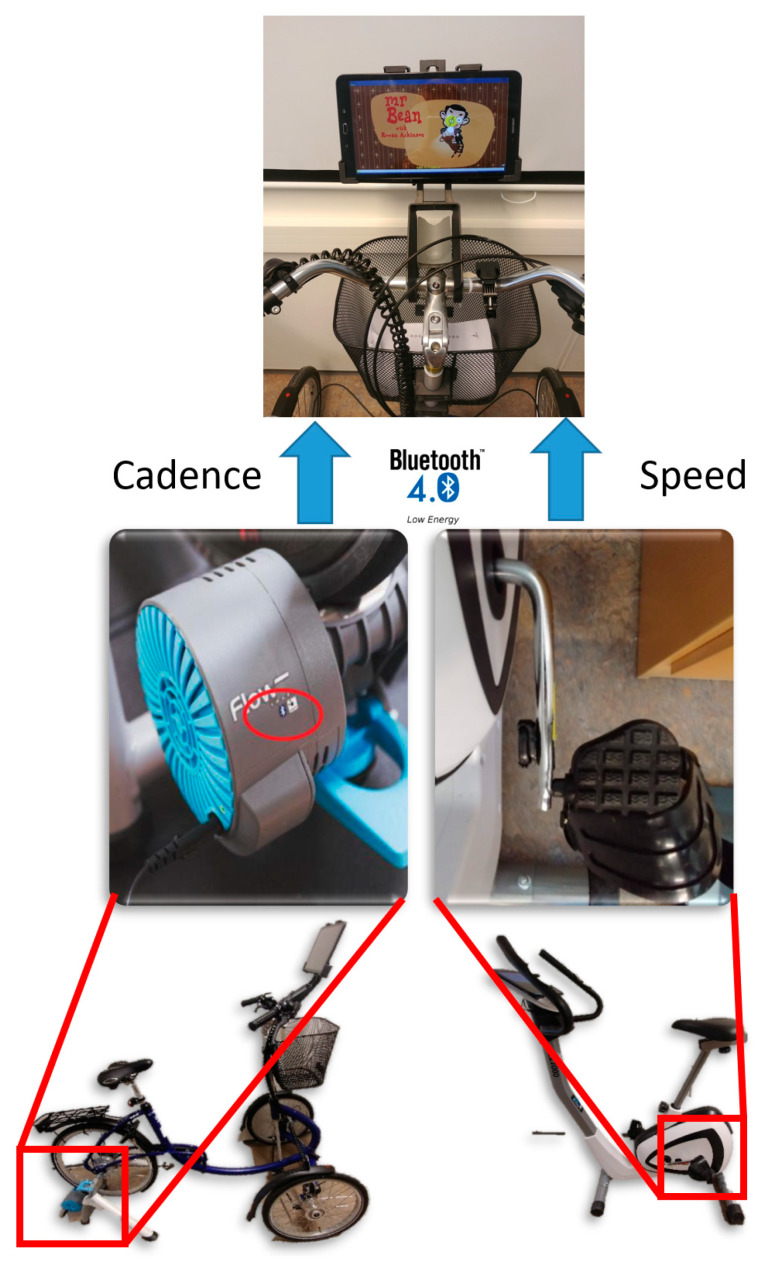
Exergame system based on static bicycle.

**Figure 3 ijerph-19-02914-f003:**
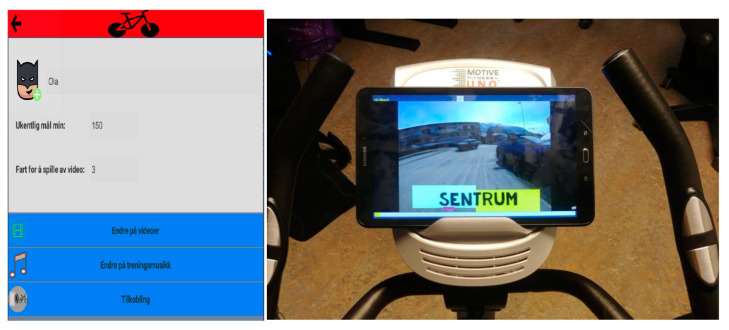
Screenshot of the app main menu (in Norwegian) and an example of a ride in the city center.

**Figure 4 ijerph-19-02914-f004:**
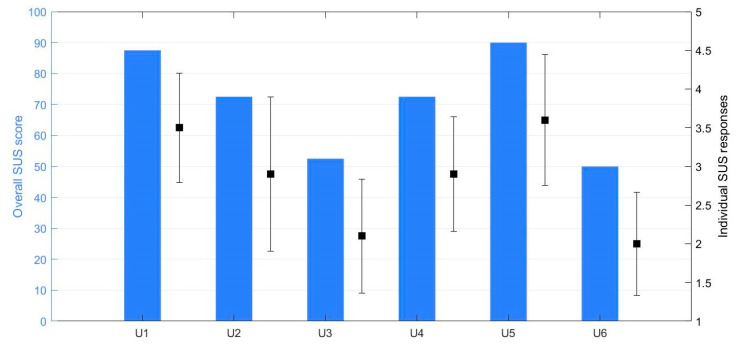
SUS scores for each testing user (blue) and the scatter of response variability (black) for the supervision group.

**Table 1 ijerph-19-02914-t001:** Description of the use cases of the system.

**Use Case 1**	
Persona	Per is a 17-year-old adolescent with Down syndrome who lives with his parents.
Preferences	Per is not very fond of being active and would instead like to lay on the couch watching Mr. Bean on his iPad. When it is summer and the weather is beautiful, Per likes to go biking with a three-wheel bike in the park, but when it is winter or the weather is bad, this is no fun.
Abilities	Per knows how to find the Mr. Bean video clips on YouTube, as he has been using this application for a long time.
Objectives	Per’s parents would like him to be more active even when it is winter or the weather is bad, so they have obtained the system and installed the app on the iPad.
Strategy	To start Per using the app, his parents had to spend some time with Per convincing him that it was a good idea, but now Per is using the bike in 10 min sessions three days a week. The system helps Per to achieve 30 min of the 150 min activity per week that is recommended by the health authorities. While Per uses the system, he likes to watch Mr. Bean and SpongeBob episodes.
**Use Case 2**	
Persona	Line is an 18-year-old adolescent that has autism and moderate intellectual disability and lives at a community residence for people with disabilities.
Preferences	In summer, Line often enjoys trips outside with her bike, but in the winter, biking outside is not recommended for her.
Abilities	The community residence that she lives in has the system, and Line uses this equipment a few times a week but she gets soon bored. She often gets confused about using gamification for practicing and fears interacting with other peers through social media.
Objectives	Staff in the community residence want Line to exercise during winter, and perform sessions of at least 30 min.
Strategy	While Line cycles on the system, she enjoys watching videos from family trips that she has been on and listening to her favorite song, which makes her feel comfortable and safe.

**Table 2 ijerph-19-02914-t002:** Key design requirements of the system tailored for persons with mild ID.

Requirement	Motivation	Criterion
The system should show progression in activity visually	Use of simple and plain visual elements for showing adequate progress and achievements	Real-time feedback on the development of physical activity with indicators related to duration and intensity
Use symbols and icons to describe navigation and actions	Few persons with mild ID are capable of reading and interpreting long texts	Ability to read should not be a requirement to use the system
Use homogeneous metaphors and animations	Predictability makes users feel comfortable and enjoy the experience	Navigation menus and elements for guiding exercise should be coherent and repetitive
The system shall play videos or music for the entertainment of the user	Individuals with ID often like to hear a song or to watch a specific movie or TV show repeatedly	A user should be able to play multimedia of his or her choice while doing physical activity
The system should have a way of setting the goal activity time and showing users the progression towards achieving this time	Persons with ID and their relatives feel motivated when rewards are given upon achievement of goals	A user should be able to set a goal time to achieve and see how far he or she has come towards this goal
The system shall not require special competence in the technology used to set up the system.	Relatives and professionals in day-care centers are rarely experts in the use of technology	A user should be able to set up the system without having special abilities in technology

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
