# Peer review of "Motivating Physical Activity for Individuals with Intellectual Disability through Indoor Bike Cycling and Exergaming"

_ijerph, 2022, doi:10.3390/ijerph19052914_

Round 1

Reviewer 1 Report

The authors have conducted a study on motivating physical activity for individuals with intellectual disability through indoor bike cycling and exergaming. They have reported the design, development and user acceptance of a mobile health solution connected to sensors to motivate physical activity. This is a meaningful research work in the field of public health.

However, the paper needs to be revised before it is considered to be accepted:

Further technical details on the implementation of the proposed method should be provided. How the research results come from needs to be explained in detail and clearly. Moreover, relevant comparative experiments need to be supplemented. More relevant new references may be included.

Author Response

A detailed report with the responses to the review is enclosed. 

Thankl you for your time

Reviewer 2 Report

In this manuscript, the authors proposed a mobile health solution aimed at stimulating physical activity to promote valuable physical activity for people with intellectual disabilities and help reduce their risk of cardiovascular disease. It is well written and makes sense for the field. However, some improvements should be made. I suggest major revision before publication as follows:

  1. The focus is not very prominent and the logic is not rigorous enough. Please rearrange this manuscript to highlight the main points.
  2. The line number of the article is slightly inconsistent with the text of the whole manuscript, such as line number "31".
  3. the whole manuscript needs re-typesetting layout, especially pictures and tables. It had better combine with the text, the overall appearance is not abrupt.
  4. the article has a large section of text description, it should supplement with related pictures, so as to make the article intuitive.
  5. The font size in the table should be adjusted to be consistent with the text font.
  6. the 7th page of the article is broken, a large white space below, expect timely adjustment.
  7. The text font should be consistent. For example, the text font under heading 3.1 is different from the previous texts.
  8. The text under heading 3.1 and the text in Table 1 are not aligned at both ends, you should adjust them appropriately.
  9. Your image quality needs to be improved. For example, you should improve the sharpness of the image in Figure 3.
  10. Read through the whole text and pay attention to the correct punctuation marks, letter capitalization and sentence grammar. For example, the end of line 29 lacks a full stop, the sentence "We Phase 2 foresees" in line 135 is grammatically incorrect, the "S" of "system" in the title of Table 1 should be capitalized "S", etc.

Author Response

(The authors gave the same response as above.)

Reviewer 3 Report

The manuscript titled "Motivating physical activity for individuals with intellectual disability through indoor bike cycling and exergaming" is an interesting research piece. The paper describe an essential understanding about the Intellectual Disability and the gamification process using principles of Exergames. The customization of the gaming for people with disabilities or people of determination is a new trend in the eSport field. In the introduction is suggested to amplify the references regarding this topic. There are several publications in the last two years related to the topic that can highlight the paper and improve the visibility of the discussion.  The methods and design are really well presented in the paper and not requires any modification. In the Ethics session. 
It is not clear if the research was submitted to a Scientific Research Ethics Committee. In the affirmative case, insert the approved report number. Another suggestion is to include two or more references from the International Journal of Environmental Research and Public Health. There are around 123 papers about gaming that can fit adequately addressing the discussion realized in the paper. The tables are a little outstanding in the paper and could be reduced bring better comprehension to the paper.  Requires also to re-write the impersonal language of the paper, removing third person elements presented in this manuscript. In general the paper is highly recommended to publish. 

Author Response

(The authors gave the same response as above.)

Round 2

Reviewer 2 Report

1. The data is too little to support your opinion except for Figure 4.

2. Table 1 and 2 need to be critically modified with much richer information (data) instead of stacking long sentences as the main text.

Author Response

Dear reviewer,

thank you for your comments and for providing us the chance to improve the paper. We understand your veredict but we are not agree with the first point. Our aim is to build a comprehensive descriptive research paper showing the methodology for user centerd design of exergames to users with intellectual disabilities - we cannot reduce our communication only to the experimental part (Figure 4) without providing an appropiate context (including advantages and limmitations. We hope you understand our views.

Regarding point number 2 we absolutelly agree, and we have improved the table to be more understandable and eye catching.

Thank you again,

The authors